# Reinfections in COVID-19 Patients: Impact of Virus Genetic Variability and Host Immunity

**DOI:** 10.3390/vaccines9101168

**Published:** 2021-10-12

**Authors:** Aisha Fakhroo, Hebah A. AlKhatib, Asmaa A. Al Thani, Hadi M. Yassine

**Affiliations:** 1Research and Development Department, Barzan Holdings, Doha 7178, Qatar; asfakhroo@barzanholdings.com; 2Biomedical Research Center, Qatar University, Doha 2713, Qatar; h.alkhatib@qu.edu.qa (H.A.A.); aaja@qu.edu.qa (A.A.A.T.)

**Keywords:** reinfection, antibodies, variants, coronavirus, SARS-CoV-2

## Abstract

The COVID-19 pandemic is still posing a devastating threat to social life and economics. Despite the modest decrease in the number of cases during September–November 2020, the number of active cases is on the rise again. This increase was associated with the emergence and spread of the new SARS-CoV-2 variants of concern (VOCs), such as the U.K. (B1.1.7), South Africa (B1.351), Brazil (P1), and Indian (B1.617.2) strains. The rapid spread of these new variants has raised concerns about the multiple waves of infections and the effectiveness of available vaccines. In this review, we discuss SARS-CoV-2 reinfection rates in previously infected and vaccinated individuals in relation to humoral responses. Overall, a limited number of reinfection cases have been reported worldwide, suggesting long protective immunity. Most reinfected patients were asymptomatic during the second episode of infection. Reinfection was attributed to several viral and/or host factors, including (i) underlying immunological comorbidities; (ii) low antibody titers due to the primary infection or vaccination; (iii) rapid decline in antibody response after infection or vaccination; and (iv) reinfection with a different SARS-CoV-2 variant/lineage. Infections after vaccination were also reported on several occasions, but mostly associated with mild or no symptoms. Overall, findings suggest that infection- and vaccine-induced immunity would protect from severe illness, with the vaccine being effective against most VOCs.

## 1. Introduction

In December 2020, new SARS-CoV-2 variants (B.1.1.7 in the United Kingdom, B.1.351 in South Africa) emerged and led to an unexpected rise in COVID-19 cases [1,2]. Both variants contained an N501Y mutation in the receptor-binding domain (RBD) of the spike (S) protein, accounting for the increased transmission and infectiousness (40–70%) of the virus [1,2]. The South African variant has two additional mutations in the RBD of the spike protein, allowing it to escape antibodies from natural infections and vaccination [3]. Taken together, due to the high transmissibility of SARS-CoV-2 along with the recent emergence of more infectious variants, the risk of reinfections may elevate. According to the Centers for Disease Control and Prevention (CDC) guidelines, SARS-CoV-2 reinfection is identified and confirmed if: (1) viral RNA is identified at two different time points; (2) intervening negative RT-PCR tests are present; and (3) viral genetic sequencing data support reinfection [4]. Several reinfection cases have been reported worldwide. The incidence rates and causes of reinfection remain poorly understood, raising many questions: Is it viral persistence (i.e., prolonged viral shedding) or reinfection? Does primary SARS-CoV-2 infection protect against subsequent infections?; How does the SARS-CoV-2 antibody response correlate with reinfection? Based on several studies, SARS-CoV-2 RNA is undetectable one month following symptom onset in the majority of patients [5,6]. Nevertheless, rare cases of prolonged viral shedding (>1 month) have been reported [6,7], including a pregnant woman who remained SARS-CoV-2-positive for 104 days after her initial test [8]. In terms of a SARS-CoV-2-specific antibody (Ab) titer, there is controversy on whether they remain stable or decline over time, although several studies reported high antibody titers that last for at least six months after infection and vaccination. In this review, we summarize the reinfection cases reported along with the humoral responses against SARS-CoV-2, with the aim of developing a better understanding of the causes of reinfection as well as the implications on vaccines.

## 2. Cases of COVID-19 Reinfection

Several cases of SARS-CoV-2 reinfections have been reported worldwide. The first reported case of reinfection occurred 4.5 months (142 days) after the first episode of COVID-19 infection [9]. A patient from Hong Kong tested positive on 26 March 2020, with mild symptoms, and was tested positive again (asymptomatic) on 15 August 2020, upon returning from Spain to Hong Kong [9]. The two episodes of infection were separated by two negative SARS-CoV-2 RT-PCR tests in April [9]. Serological testing indicated that SARS-CoV-2 IgG was negative one day following the second infection, followed by a positive test five days later. The whole-genome analysis confirmed that the SARS-CoV-2 strains from both infections belonged to different clades, including 24 nucleotide differences; of particular interest are four amino acid variants in the spike protein (L18F, A222V, D614G, and Q780E). Specifically, residues 222 and 614 are positioned within proximity to previously identified B cell epitopes, hence possibly affecting antibody response. Based on the genomic analysis, the first strain was closely related to U.S. strains, whereas the second strain closely resembled strains from Switzerland and England, which reinforces the idea of reinfection given that the patient traveled back from Europe [9]. In a larger cohort study analyzing reinfection rates in Qatar, most of the patients were asymptomatic or experienced mild symptoms after reinfection [10]. In this study, which included 133,266 laboratory-confirmed SARS-CoV-2 cases, 54 out of 243 positive cases (22.2%) showed a high probability of reinfection (i.e., tested positive ≥45 days following the first positive test), out of which four reinfections were confirmed through viral genome sequencing [10]. In another study from Qatar, it was reported that SARS-CoV-2 antibody positivity protects from reinfection for at least seven months with 95% efficacy [11]. In this study, 43,044 antibody-positive persons were followed for a median of 16.3 weeks (range: 0–34.6), of which 314 individuals (0.7%) had at least one positive PCR swab ≥14 days after the first positive antibody test. Of the positive individuals, 129 (41.1%) had supporting epidemiological evidence for reinfection. Reinfection was analyzed using viral genome sequencing. Applying this methodology to confirm reinfection, the incidence rate of reinfection was estimated at 0.66 per 10,000 person-weeks (95% CI: 0.56–0.78). In another study, a 51-year old woman experienced a relapse of symptoms with no change in severity within three months after the first infection [12]. The full-length genome sequencing revealed that the viral genome from the first and second infection belonged to lineage B.1.1 and lineage A, respectively, with 11 mutations identified [12]. On the other hand, some cases of reinfection reported an increase in disease severity [13,14]. The first case of reinfection in North America reported an increase in symptom severity during the second infection; a 25-year-old man relapsed and was tested positive six weeks after the first episode of infection, separated by two negative RT-PCR tests [13]. During the reinfection, the patient tested positive for IgG and IgM against SARS-CoV-2; however, his immune reaction was not assessed during the first episode of infection, thus the duration and degree of immunity were inconclusive [13]. Sequencing data revealed that the viral genome from both infections belonged to the same clade (20 C), with multiple SNV (single-nucleotide variant) and MNV (multiple-nucleotide variant) mutations, raising the possibility of continuous viral infection [13]. Nevertheless, the odds of infection reactivation are quite low; the mutation rate (i.e., rate of SNV and MNV accumulation) for the two viral genomes was calculated to be 83.64 substitutions per year, which significantly exceeds the current reported rate of 23.12 [13]. Similarly, an 89-year-old Dutch woman relapsed 59 days after the first episode of infection with an increase in symptom severity: the patient’s condition deteriorated, and she died two weeks following the second infection [14]. However, the patient was immunocompromised in this case, suffering from Waldenström macroglobulinemia and undergoing B cell-depleting therapy [14]. Antibodies against SARS-CoV-2 were tested negative, and the two strains varied at ten nucleotide positions in ORF1a (4), ORF1b (2), spike (2), ORF3a (1), and M (1) genes [14]. Despite the lack of PCR-negative samples in between the two infections, the second infection was likely to be reinfection rather than continuous viral shedding due to the average estimated SARS-CoV-2 mutation rate of 33 nucleotides per year [14]. In a Brazilian study describing the recurrence of COVID-19 in 33 patients, the recurrent episodes also tended to be more severe than the first episode, including one lethal infection [15]. However, only one patient was verified as reinfection through genomic sequencing, where the sequences from the first and second symptomatic episodes belonged to different clades (Lineages B.1and B.1.80) [15].

Several risk factors are associated with COVID-19 reinfection, such as being a healthcare worker, a blood-group A person, or having low antibody (IgG) titers [15]. In addition, a newly classified P2 variant harboring the E484K mutation has been detected in two SARS-CoV-2 reinfection cases in Brazil, which raises concern regarding the potential impact of this mutation on immune escape [16,17,18]. In terms of disease severity associated with the P2 variant, one case exhibited mild symptoms during reinfection [16], whereas the other exhibited more severe symptoms compared to the first round of infection [17]. It is worth noting that most of the patients that experienced milder symptoms during reinfection exhibited lower antibody titers, partially explaining the higher possibility for reinfection. Previous studies showed that COVID-19 severity positively correlates with antibody response. In other words, patients with milder symptoms exhibit lower antibody titers than severely ill patients [19,20]. Upon studying 37 asymptomatic individuals, 40% were found seronegative within eight weeks after testing positive, although exhibiting longer RNA shedding compared to severe cases as well [19,20]. Collectively, these data indicate that the majority of reinfections are asymptomatic, suggesting sufficient immunological memory. The cases are summarized in Table 1.

## 3. Humoral Response against SARS-CoV-2

The duration and dynamics of humoral Ab responses against SARS-CoV-2 remain poorly understood. Acute antibody responses were recorded in 285 COVID-19 patients, where 100% of the patients tested positive for SARS-CoV-2 IgG within 19 days following symptom onset [21]. Meanwhile, both IgG and IgM titers stabilized (i.e., reached a plateau) within six days following seroconversion [21]. In another study, including 173 COVID-19 patients, the seroconversion rates of total antibodies, IgM, and IgG were found to be 93.1%, 82.7%, and 64.7%, respectively, with average seroconversion durations of 11, 12, and 4 days, respectively [22]. The levels of antibodies were quite low (<40%) within one week after symptom onset, followed by a rapid increase to 100.0% (Ab), 94.3% (IgM), and 79.8% (IgG) by day 15 [22]. Additionally, an association between high antibody titers and severe clinical manifestations was observed [22]. Similarly, among 52 COVID-19 patients, the median seroconversion times were eight days (44/52, 84.6%) and ten days (42/52, 80.8%) for IgM and IgG, respectively, with significantly higher titers recorded for ICU (Intensive Care Unit) patients compared to those with milder disease [23]. In terms of antibody response duration and stability, a longitudinal seroprevalence study of 3276 U.K. healthcare workers (HCWs) recorded SARS-CoV-2 anti-nucleocapsid and anti-spike IgG levels over six months [24]. Anti-spike IgG was still stably detectable within 180 days after symptom onset in 94% of the seropositive HCWs, whereas anti-nucleocapsid IgG levels peaked at 24 days post-onset followed by a drop over time, with an average calculated half-life of 85 days [24]. Furthermore, a faster decline of anti-nucleocapsid IgG levels was observed in younger adults and asymptomatic cases [24]. In another longitudinal cohort study including U.K. HCWs, a low number of reinfections was reported in seropositive HCWs over six months of follow-up [25]. Thereby, previous infection resulting in anti-spike or anti-nucleocapsid IgG antibodies was correlated with a significantly reduced risk of SARS-CoV-2 reinfection following six months [25]. Studies from Iceland and the United States indicated that SARS-CoV-2 antibody levels did not decline and remained relatively stable within four months post-infection [26,27]. In contrast, other studies reported a rapid decline in antibody levels within 3–4 months post-infection [28,29,30]. Among 27 COVID-19 convalescent patients, all except one (#10) experienced a significant decline in SARS-CoV-2-specific IgG and IgA, 100 days post-hospital discharge. Meanwhile, a significant reduction in SARS-CoV-2-specific IgM was observed in all patients except one (#14) within 100 days after hospital discharge [30]. In summary, the majority of COVID-19 patients elicit detectable an antibody response within 10–14 days post-infection, with milder cases displaying lower antibody titers than severe cases. The difference in antibody longevity between the different studies could be attributed to the type of tests or methodologies used, knowing that anti-S IgG levels seem to last longer than anti-nucleocapsid IgG levels [24]. Additionally, humoral response dynamics and longevity have been shown to vary at the individual level [31]. It is worth noting that a rapid decline in antibody titers was reported within one year after infection with MERS-CoV, especially in asymptomatic or mildly symptomatic patients [32].

## 4. Possible Causes of Reinfections

Based on the several reported SARS-CoV-2 reinfection cases, primary exposure to SARS-CoV-2 does not necessarily confer protection from reinfection. In contrast, animal studies in rhesus macaques indicated that infection with SARS-CoV-2 resulted in immune responses that protect against subsequent reinfection [33,34]. Therefore, understanding the causes of reinfections requires a better understanding of the nature, duration, and dynamics of the anti-SARS-CoV-2 antibody response, as well as the variables associated with the virus itself. Firstly, the second round of infection could be a result of exposure to a higher dose of the virus, thus leading to reinfection or a more severe disease [35]. Secondly, the low antibody titers reported in asymptomatic/mild cases [19,20,36] may correlate with a high chance of reinfection. It is worth noting here that a positive PCR test, especially at higher CT values (>30), does not indicate an active infection, because this could be a transient exposure to the virus that does not establish an actual infection. A U.K. study observed a strong connection between CT value and viral infectivity; the probability to culture an infectious virus decreased to 8% in patients with CT > 35 [37]. In correspondence, a study established a correlation between low NAb (neutralizing antibody) titers and SARS-CoV-2 reinfection using a ferret reinfection model [38]. Unlike the low NAb titer group, rapid viral clearance and reduced viral replication were observed in the high NAb titer group [38]. Additionally, only the low NAb titer group (<20) exhibited direct-contact transmission [38]. Furthermore, the recorded antibody decline in several studies [28,29,30] may explain reinfection. Thirdly, according to the reinfection cases reported, the viral genomes from both infections either belonged to different clades, lineages or differed by a considerable number of mutations (>2 substitutions per month) [9,10,12,13,14]. Therefore, the reinfection may be caused by a SARS-CoV-2 version that is more virulent/infectious/transmissible, including mutations that allow them to evade the host’s immune response [13]. In this regard, studies have shown up to a tenfold drop in neutralization activity against the South African and Brazilian variants in individuals receiving mRNA vaccines using the original strain [39,40]. Moreover, the immunological comorbidities associated with some patients may induce reinfection, which potentially accounts for the reinfection and symptom relapses documented in an 89-year-old immunocompromised woman [14]. Finally, antibody-dependent enhancement (ADE) may trigger reinfection, in which the virus utilizes Fc-bearing immune cells and binds to the antibodies, enhancing viral entry into host cells [41]. The closely related SARS-CoV-1 has previously been shown to undergo this mechanism [41]. In the case of reinfection with strains from the same clade, co-infection should not be ruled out [13]. In such a condition, the patient could be infected with two different genotypes of the virus (e.g., Genome A and Genome B) at the same time, although with different quantities. Particularly, genome A was detectable whereas genome B was not during the first infection and vice versa; genome A depleted, and genome B is now detectable during the second onset [13]. In those reinfection cases [9,10,12,13,14], within-host evolution is ruled out because for it to be applicable, the virus’s mutation rate would need to highly exceed the current recorded rate of 23.12 [13]. Another possible explanation for such cases is viral deactivation/reactivation. A recent study proposed the potential role of exosomes in SARS-CoV-2 reactivation, where the viral genome hides within vesicles and is reactivated within 7–14 days post-infection; a mechanism that has been described for other viruses [42]. However, all of the reinfection durations reported exceed 14 days, which makes this hypothesis not applicable [9,10,12,13,14]. Overall, several viral and/or host factors account for the possibility of reinfection (Figure 1).

## 5. Vaccination and Protection from Infection

Several vaccine platforms have been licensed for emergency use worldwide, most of which provide a high degree of protection from disease and hospitalization [43]. However, the degree of protection from infection is still controversial. In a study investigating the correlation between previous SARS-CoV-2 infection and BNT162b2 vaccine-induced immune responses (i.e., humoral and T cell responses), individuals with prior exposure to SARS-CoV-2 exhibited more robust humoral and cellular responses compared to infection-naïve individuals (i.e., not infected previously with SARS-CoV-2) [44]. Upon administrating one dose of the vaccine, individuals with previous SARS-CoV-2 infection generated significantly higher anti-S titers than infection-naïve individuals, which potentially implies that one dose may be sufficient for those previously infected [44]. The T cell responses against spike peptides were also significantly weaker (negative in 50% of the participants) in infection-naïve individuals than in individuals with previous exposure [44]. Additionally, an inverse correlation between post-vaccination anti-S titers and age was observed in infection-naïve individuals, where individuals >50 years old produced a weaker humoral response than those <50 years old [44]. Of note, one of the infection-naïve individuals developed symptomatic COVID-19 five weeks following the first dose of vaccine despite generating an anti-S titer of 61.8 AU/mL [44]. Similarly, several cases of COVID-19 have been reported in individuals who received one or two doses of the mRNA vaccine [45,46]. In a cohort study including 36,659 HCWs from the University of California, San Diego, CA, USA (UCSD) and the University of California, Los Angeles, CA, USA (UCLA), 28,184 (77%) proceeded and received a second dose of the vaccine [46]. Among the vaccinated HCWs, a positive SARS-CoV-2 test was observed in 379 individuals as early as one day post-vaccination [46], of which, 71% tested positive within two weeks after the first dose, and 37 tested positive after the second dose [46]. The time of infection following the second dose varied: 22 positive cases were reported within 1–7 days, eight positive cases were reported within 8–14 days, and eight positive cases were reported 15 days or more after the second dose [46]. In general, the absolute risk of SARS-CoV-2 incidence post-vaccination was calculated to be 1.19% and 0.097% in UCSD and UCLA, respectively, which exceeds the risk rates reported in the trials of mRNA-1273 and BNT162b2 vaccines [46]. According to a CDC report, more than 95 million people in the United States were vaccinated against COVID-19 as of 2 May 2021, including a total of 9245 vaccine breakthrough infections with hospitalization and fatality rates of 9% and 1%, respectively [47]. Interestingly, 63% of CDC-reported vaccine breakthrough infections were female [34]. Furthermore, the emergence of new variants of concern (VOCs) poses an imminent threat to vaccine campaigns. Using a mathematical modeling approach, one study showed that our current strategy to control COVID-19 (i.e., rapid vaccination along with partial lockdown) would minimize the spread of highly contagious SARS-CoV-2 variants [35]. However, this study took into consideration the B.1.1.7 variant only [35]. In a study from Qatar, the estimated effectiveness of BNT162b2 vaccines against B.1.1.7 and B.1.351 variants was found to be 89.5% and 75.0%, respectively [48]. Nevertheless, the effectiveness of the vaccine against severe SARS-CoV-2 infection (including both variants) was very high, reaching 97.4%, indicating vaccine protection from increased severe illness [48,49]. Similarly, vaccination and prior infection were associated with lower risks of a SARS-CoV-2-positive PCR test in residents returning to Qatar on international flights [50]. However, the authors indicated that both vaccine immunity and natural immunity were imperfect, with breakthrough infections recorded. Interestingly, out of 72 positive PCR cases, the strains identified were B.1.351 (beta; *n* = 32; 44.4%), B.1.1.7 (alpha; *n* = 20; 27.8%), B.1.617 (delta; *n* = 8; 11.1%), and “wild-type” strains (*n* = 12; 16.7%) [50].

There is a concern that the vaccine might not be effective against some other variants, specifically B.1.351, P.1, and B.1.617. These three VOCs were shown to escape the antibodies induced by infection and vaccination [36]. They harbor several mutations in the spike protein, including a key mutation called E484K/E484Q which probably confers resistance to antibodies, aiding in high transmissibility [36]. In relation, in a cohort study of 417 fully vaccinated individuals (Pfizer or Moderna), two women tested positive for SARS-CoV-2 post-vaccination [51]. Through viral sequencing, these vaccine breakthrough infections were shown to be associated with E484K mutation in one woman and three other mutations (T95I, del142–144, and D614G) in both [51]. Therefore, these data reinforce the potential risk of infections after vaccination, particularly with emerging VOCs. Additionally, the rapidly growing variant (B.1.427/B.1.429) in California (i.e., up to >50% of cases by early 2021) was shown to be 20% more transmissible, with twofold increased shedding in vivo [52]. Similarly to the B.1.351 and P.1 variants, the antibody neutralization was also reduced with the “California” variant, leading to lower protection rates following vaccination [52]. The “California” variant contains a key mutation in its spike (L452R), leading to increased infectivity in vitro [52]. This mutation is also found in the “Indian” variant (B.1.617), which contributed to the rapid rise in coronavirus cases and deaths in India. Infection cases post-vaccination are being reported and new SARS-CoV-2 variants with increase infectivity are emerging [44,47,51,52]; therefore, the possibility of reinfections post-vaccination should also be investigated (i.e., whether vaccination protects from reinfection). Moreover, this raises an important question as to whether those proven to be infected twice with SARS-CoV-2 are eligible for vaccination or not, given that they already exhibit high antibody titers.

## 6. Conclusions

In conclusion, this review summarizes cases of SARS-CoV-2 reinfection with an emphasis on its relationship to the humoral immune response. Thoroughly studying the immune response will help us better understand the causes of reinfection. Despite the limited number of reinfection cases reported worldwide, reinfection may be more common but poorly detected (i.e., missed) due to (1) lack of RT-PCR tests post-infection, (2) lack of serological testing, or (3) people not reporting symptoms with the misassumption that primary SARS-CoV-2 infection provides complete protection. Accordingly, more longitudinal studies should be conducted while taking into consideration possible reinfection implications on the vaccines, hence providing an insight into the eligibility of different groups for vaccination.

## Figures and Tables

**Figure 1 vaccines-09-01168-f001:**
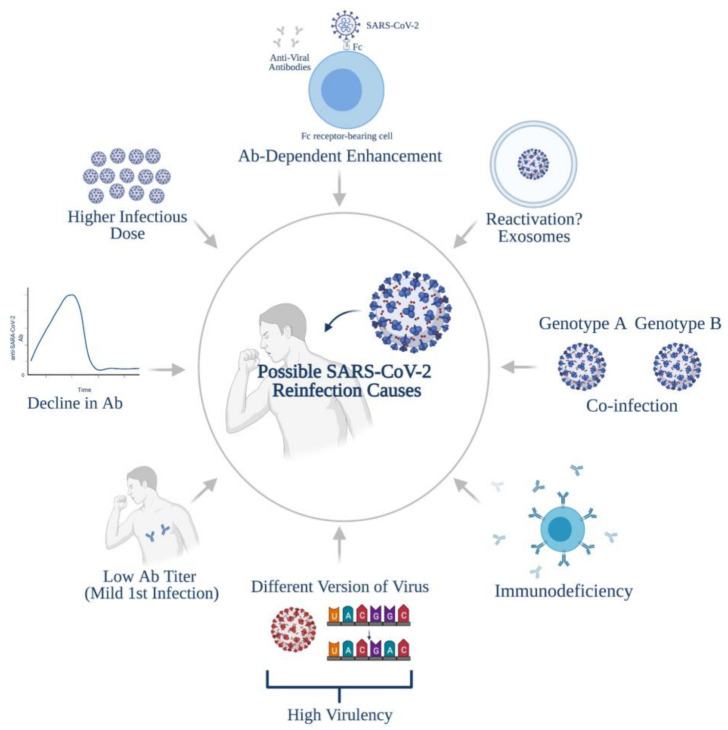
Possible causes of SARS-CoV-2 reinfections. Created using BioRender.com.

**Table 1 vaccines-09-01168-t001:** Brief summary of some SARS-CoV-2 reinfection cases.

Study	Country	Duration between Infections	Immunological Comorbidities	Severity	
				First Episode of Infection	Second Episode of Infection
[9]	Hong Kong	142 days	No	Mild	Asymptomatic
[10]	Qatar	46 days	No	Mild/Asymptomatic	Mild/Asymptomatic
[10]	Qatar	71 days	No	Mild/Asymptomatic	Mild/Asymptomatic
[10]	Qatar	88 days	No	Mild/Asymptomatic	Mild/Asymptomatic
[10]	Qatar	55 days	No	Mild/Asymptomatic	Mild/Asymptomatic
[12]	Belgium	3 months	No	Mild	Mild
[13]	USA	42 days	No	Mild	Severe
[14]	Netherlands	59 days	Yes (Waldenström macroglobulinemia)	Severe	Severe (death)
[15]	Brazil	52 days	No	Mild–Severe	Mild
[16]	Brazil	116 days	No	Mild	Mild
[17]	Brazil	147 days	No	Mild	Severe

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
