# Peer review of "Reinfections in COVID-19 Patients: Impact of Virus Genetic Variability and Host Immunity"

_vaccines, 2021, doi:10.3390/vaccines9101168_

Round 1

Reviewer 1 Report

The review by Fakhroo et al. is a useful and fairly presented synopsis of a rapidly changing set of data about COVID-10 reinfections.  It offers brief statements about some of the biological and technical complexities that may explain results, including discordances between studies; a good example of this is in Section 4 with the parenthetical observation of virus possibly "more virulent (i.e., generally or in the patient's context)", this being a subtlety inapparent to most.  In Section 5, a careless reader could infer vaccine failures occurring at a much higher rate than currently known; beware unintentional fueling of vaccine hesitancy.  There are a few scattered editorial things to catch (e.g. verb tense disagreements, and typos), and the last two citations (37 & 38) are missing from the list of references.

Author Response

Reviewer 1:

The review by Fakhroo et al. is a useful and fairly presented synopsis of a rapidly changing set of data about COVID-10 reinfections.  It offers brief statements about some of the biological and technical complexities that may explain results, including discordances between studies; a good example of this is in Section 4 with the parenthetical observation of virus possibly "more virulent (i.e., generally or in the patient's context)", this being a subtlety inapparent to most.  In Section 5, a careless reader could infer vaccine failures occurring at a much higher rate than currently known; beware unintentional fueling of vaccine hesitancy.  There are a few scattered editorial things to catch (e.g. verb tense disagreements, and typos), and the last two citations (37 & 38) are missing from the list of references.

Response: Thank you for the positive feedback, it is highly appreciated. In section 5, additional studies [47-49] highlighting vaccine efficiency have been discussed (p.7, highlighted in yellow) to avoid vaccine hesitancy. Apologies for missing out the last two citations, references 37 and 38 have been added. The document was screened through software Grammarly to fix the typo/English mistakes.

Reviewer 2 Report

The review article written by Fakhroo et al., on "Reinfections in COVID-19 patients: Impact of virus genetic variability and host immunity" describes reinfection and its relation to antibody responses and strain variations. The article covers all related literature on Covid-19 reinfection and possible speculations on reinfection. However, still, need some improvement in terms of rewriting and adding additional references upon agreeing to make it better

  1. Abstract - Overall, a limited number of reinfection cases were reported worldwide, suggesting a long protective immunity. did author mean long protective immunity achieved by both vaccines and natural infections?
  2. it is not very clear 22.2% "In this study that included 133,266 laboratory-confirmed SARS-CoV-2 cases, 54 cases (22.2%) showed a high probability of reinfection (10)". Mention that this study has 133,266 participants and out 243 positives, 54 reinfection that gives 22.2%".
  3.  spell check/corrections needed - For eg page 3 - macrogbulinemia; fig 1 - Fc bearing cells? cells bear FcgReceptors

  4.  

    The Author could consider discussing the role of residual antibodies from primary infection during reinfection/ the role of T-cell memory/impact of non-spike variants which is relevant for this review work.

  5.  

    Also consider including the recent work "Sabino, E. C. et al. Resurgence of COVID-19 in Manaus, Brazil, despite high seroprevalence. Lancet 397, 452–455 (2021)" that shows strong evidence of reinfection even after 6-8 months due to waning of immune response

     Author Response

The review article written by Fakhroo et al., on "Reinfections in COVID-19 patients: Impact of virus genetic variability and host immunity" describes reinfection and its relation to antibody responses and strain variations. The article covers all related literature on Covid-19 reinfection and possible speculations on reinfection. However, still, need some improvement in terms of rewriting and adding additional references upon agreeing to make it better

Response: Thank you pointing out the areas to improve, we acknowledge the thorough comments, which were addressed accordingly.

Note: All changes are highlighted or track changed.

  1. Abstract - Overall, a limited number of reinfection cases were reported worldwide, suggesting a long protective immunity. did author mean long protective immunity achieved by both vaccines and natural infections?

Response: Several studies have now reported protection from severe disease after vaccination and reinfection. We highlighted new studies from Qatar and other countries in the revised manuscript.

  1. it is not very clear 22.2% "In this study that included 133,266 laboratory-confirmed SARS-CoV-2 cases, 54 cases (22.2%) showed a high probability of reinfection (10)". Mention that this study has 133,266 participants and out 243 positives, 54 reinfection that gives 22.2%".

Response: Thank you for pointing out the mistake, it was adjusted (P.2).

  1.  spell check/corrections needed - For eg page 3 - macrogbulinemia; fig 1 - Fc bearing cells? cells bear FcgReceptors

Response: Apologies, the spelling mistakes were adjusted accordingly (P.3 and P.6).

  1.  The Author could consider discussing the role of residual antibodies from primary infection during reinfection/ the role of T-cell memory/impact of non-spike variants which is relevant for this review work.

Response: Thank you for the suggestion. The role of residual antibodies from primary infection were discussed briefly under section 4 (Possible Pauses of Reinfection, P.4-5), study [37] on the correlation between low NAb and and SARS-CoV-2 reinfection in recovered ferret reinfection model.

  1.  Also consider including the recent work "Sabino, E. C. et al. Resurgence of COVID-19 in Manaus, Brazil, despite high seroprevalence. Lancet 397, 452–455 (2021)" that shows strong evidence of reinfection even after 6-8 months due to waning of immune response

Response: Thank you for the suggestion. This study [18] was added in section 2 (Cases of COVID-19 reinfection).

Reviewer 3 Report

The work by Fakhroo et al. summarizes the reasons behind SARS-CoV-2 reinfections in previously infected and vaccinated individuals. The review was broken down into cases of reinfections, modes of reinfection, and the role of humoral responses and vaccination in preventing reinfection. In general, the review is interesting and potentially important for the field. I have some comments for the authors to consider.

  1. The authors have not cited some important papers in their review, kindly introduce them.
    • Humoral responses section: Cochrane Database Syst Rev. 2020 Jun 25;6(6):CD013652. &  N Engl J Med. 2021 Feb 11;384(6):533-540. 
    • Possible Cause of Reinfections: Viruses. 2021 Apr 16;13(4):697, Emerging Microbes & Infections, 10:1, 152-160, mBio. 2021 Jan 19;12(1):e02940-20
  2. References need to be formatted as per the journal guideline. It was very difficult to find some of the papers cited in the paper due to incorrect formatting. 
  3. References 37 & 38 are missing from the Reference section 

Author Response

Reviewer 3:

The work by Fakhroo et al. summarizes the reasons behind SARS-CoV-2 reinfections in previously infected and vaccinated individuals. The review was broken down into cases of reinfections, modes of reinfection, and the role of humoral responses and vaccination in preventing reinfection. In general, the review is interesting and potentially important for the field. I have some comments for the authors to consider.

Response: We appreciate the thorough and positive feedback.

Note: All changes are highlighted or track changed.

  1. The authors have not cited some important papers in their review, kindly introduce them.
    • Humoral responses section: Cochrane Database Syst Rev. 2020 Jun 25;6(6):CD013652. &  N Engl J Med. 2021 Feb 11;384(6):533-540. 
    • Possible Cause of Reinfections: Viruses. 2021 Apr 16;13(4):697, Emerging Microbes & Infections, 10:1, 152-160, mBio. 2021 Jan 19;12(1):e02940-20

Response: Thank you for suggesting these papers. We have read through the papers and we believe that NEJM, Viruses, Microbes and Infections papers are of relevance to this review. Hence, we referenced NEJM paper [25] in P.4 under section 3 (Humoral response against SARS-CoV-2). The other two papers [35, 37] were discussed in P.4-5 under section 4 (Possible Causes of Reinfections).

  1. References need to be formatted as per the journal guideline. It was very difficult to find some of the papers cited in the paper due to incorrect formatting. 

Response: Apologies, references were formatted according to the journal guideline.

  1. References 37 & 38 are missing from the Reference section 

Response: Apologies for missing out the last two citations, references 37 and 38 have been added.

Reviewer 4 Report

In their manuscript entitled „Reinfections in COVID-19 patients: Impact of virus genetic varability and host immunity” Fakhroo and colleagues present an overview of potentially causes of reinfection with SARS-CoV-2. Moreover, they relate reinfection to the antibody response. The manuscript is well structured and clearly written. Although reinfection seems not to be very common, there is already a lot of literature out there regarding this topic and also humoral response and protection. Hence, for a review article, I strongly recommend to perform a deeper literature research and support the review article with more than the 36 cited references. The most important points in my opinion are the following:

  • Page 2, section „2. Cases of COVID-19 Reinfection”. The references should be extended to conclude “that (the) majority of reinfections are asymptomatic,”.
  • Table 1: 4/9 cases are from Qatar. The authors should consider a more balanced and comprehensive case overview in a review article.
  • On page 4 the authors state:” (…) the second round of infection could be a result of exposure to a higher dose of the virus, thus leading to a severer disease [29].” Could a higher infection dose itself lead to reinfection in general (not necessarily more severe)?
  • On page 4 the authors write: “It is worth noting here that a positive PCR-test, especially at higher CT-values (>30), does not indicate a positive infection, as that could be a transient exposure to the virus that does not establish an actual infection.” The cT value strongly depends on the PCR protocol and previous sample preparation steps. It could be speculated that high cT values do not correspond to an acute infection. However, I am not aware of any reference supporting this statement. Hence, the authors should to support this statement with according references or state that this is a speculation. Additionally, PCR tests can be false positive, which is another point to consider when talking about reinfection.
  • Page 4: “In this regard, studies have shown about a ten-fold drop in neutralization activity against the South African and Brazilian variants in individuals receiving mRNA vaccine using the original strain.” The statement is missing a reference.
  • Chapter 3” 3. Humoral Response against SARS-CoV-2”, page 4: The statement “The difference in antibodies longetivity between the different studies could be attributed to the type of tests or methodologies used, know-ing that anti-S IgG levels seem to last longer that anti-nucleaocapsid IgG levels.” (1.) The statement is missing a reference and (2.) I think that the different tests should be considered, but I think that they are just a minor issue here. It has been shown that the humoral response is quite individual, e.g. in the study from Wan Ni Chia and colleagues (Dynamics of SARS-CoV-2 neutralising antibody responses and duration of immunity: a longitudinal study)
  • Figure 1: The figure is in some points misleading. I recommend the following changes: (1.) Replace “high viral titer” with “high infectious dose” (2.) Make the difference between “different versions of virus” and the co-infection with different versions clearer.

Author Response

Reviewer 4:

In their manuscript entitled “Reinfections in COVID-19 patients: Impact of virus genetic varability and host immunity” Fakhroo and colleagues present an overview of potentially causes of reinfection with SARS-CoV-2. Moreover, they relate reinfection to the antibody response. The manuscript is well structured and clearly written. Although reinfection seems not to be very common, there is already a lot of literature out there regarding this topic and also humoral response and protection. Hence, for a review article, I strongly recommend to perform a deeper literature research and support the review article with more than the 36 cited references. The most important points in my opinion are the following:

Response: Thank you for the thourough feedback and comments, highly appreciated. Deeper literature research was performed, including more studies, which expanded the references to a total of 52 references.

Note: All changes are highlighted or track changed.

  • Page 2, section„2. Cases of COVID-19 Reinfection”. The references should be extended to conclude “that (the) majority of reinfections are asymptomatic,”.

Response: The references were summarized to conclude that the majority of reinfections are asymptomatic.

  • Table 1: 4/9 cases are from Qatar. The authors should consider a more balanced and comprehensive case overview in a review article.

Response: Thank you for pointing this out. Additional studies [16-18] on reinfection cases from Brazil were added to section 2 (Cases of COVID-19 reinfections), which therefore expanded Table 1.

  • On page 4 the authors state:” (…) the second round of infection could be a result of exposure to a higher dose of the virus, thus leading to a severer disease [29].” Could a higher infection dose itself lead to reinfection in general (not necessarily more severe)?

Response: It could lead to reinfection in general and in some cases to more severe disease. This sentence was rephrased accordingly and added to P.4, section 4 (Possible causes of reinfections).

  • On page 4 the authors write: “It is worth noting here that a positive PCR-test, especially at higher CT-values (>30), does not indicate a positive infection, as that could be a transient exposure to the virus that does not establish an actual infection.” The cT value strongly depends on the PCR protocol and previous sample preparation steps. It could be speculated that high cT values do not correspond to an acute infection. However, I am not aware of any reference supporting this statement. Hence, the authors should to support this statement with according references or state that this is a speculation. Additionally, PCR tests can be false positive, which is another point to consider when talking about reinfection.

Response: Thank you for pointing this out. This point was taken into consideration, hence study [36] was added to support the statement. This study showed that the probability to culture infectious virus decreased to 8% in patients with CT>35.

  • Page 4: “In this regard, studies have shown about a ten-fold drop in neutralization activity against the South African and Brazilian variants in individuals receiving mRNA vaccine using the original strain.” The statement is missing a reference.

Response: Apologies for the missing references, references [38,38] were added accordingly to support the statement.

  • Chapter 3” 3. Humoral Response against SARS-CoV-2”, page 4: The statement “The difference in antibodies longetivity between the different studies could be attributed to the type of tests or methodologies used, know-ing that anti-S IgG levels seem to last longer that anti-nucleaocapsid IgG levels.” (1.) The statement is missing a reference and (2.) I think that the different tests should be considered, but I think that they are just a minor issue here. It has been shown that the humoral response is quite individual, e.g. in the study from Wan Ni Chia and colleagues (Dynamics of SARS-CoV-2 neutralising antibody responses and duration of immunity: a longitudinal study)

Response: Apologies, reference [24] was added to support the statement. Also, Thank you for the suggestion, we added the study from Wan Ni Chia and colleagues [31], which shows that SARS-CoV-2 humoral responses vary at the individual level.

  • Figure 1: The figure is in some points misleading. I recommend the following changes: (1.) Replace “high viral titer” with “high infectious dose” (2.) Make the difference between “different versions of virus” and the co-infection with different versions clearer.

Response: Changes were applied to figure 1 accordingly. (1) “high viral titer” was replaced with “high infectious dose” and (2) to make “co-infection” clearer, “Version A and B” were replaced by “Genotype A and B”, indicating they are different genotypes from the same virus (same clade).